# Novel Small Molecule Positive Allosteric Modulator SPAM1 Triggers the Nuclear Translocation of PAC1-R to Exert Neuroprotective Effects through Neuron-Restrictive Silencer Factor

**DOI:** 10.3390/ijms232415996

**Published:** 2022-12-15

**Authors:** Guangchun Fan, Shang Chen, Lili Liang, Huahua Zhang, Rongjie Yu

**Affiliations:** 1Department of Cell Biology, College of Life Science and Technology, Jinan University, Guangzhou 510632, China; 2Department of Medical Genetics, Guangdong Medical University, Dongguan 523808, China; 3Guangdong Province Key Laboratory of Bioengineering Medicine, Guangzhou 510632, China; 4Guangdong Provincial Biotechnology Drug & Engineering Technology Research Center, Guangzhou 510632, China; 5National Engineering Research Center of Genetic Medicine, Guangzhou 510632, China

**Keywords:** neuropeptide pituitary adenylate cyclase-activating polypeptide (PACAP), PAC1-R, positive allosteric modulator (PAM), SPAM1, D-galactose (D-gal), huntingtin, neuroprotective effect, neuron-restrictive silencer factor (NRSF)

## Abstract

The neuropeptide pituitary adenylate cyclase-activating polypeptide (PACAP) exerts effective neuroprotective activity through its specific receptor, PAC1-R. We accidentally discovered that as a positive allosteric modulator (PAM) of PAC1-R, the small-molecule PAM (SPAM1) has a hydrazide-like structure, but different binding characteristics, from hydrazide for the N-terminal extracellular domain of PAC1-R (PAC1-R-EC1). SPAM1 had a significant neuroprotective effect against oxidative stress, both in a cell model treated with hydrogen peroxide (H_2_O_2_) and an aging mouse model induced by D-galactose (D-gal). SPAM1 was found to block the decrease in PACAP levels in brain tissues induced by D-gal and significantly induced the nuclear translocation of PAC1-R in PAC1R-CHO cells and mouse retinal ganglion cells. Nuclear PAC1-R was subjected to fragmentation and the nuclear 35 kDa, but not the 15 kDa fragments, of PAC1-R interacted with SP1 to upregulate the expression of Huntingtin (Htt), which then exerted a neuroprotective effect by attenuating the binding availability of the neuron-restrictive silencer factor (NRSF) to the neuron-restrictive silencer element (NRSE). This resulted in an upregulation of the expression of NRSF-related neuropeptides, including PACAP, the brain-derived neurotrophic factor (BDNF), tyrosine hydroxylase (TH), and synapsin-1 (SYN1). The novel mechanism reported in this study indicates that SPAM1 has potential use as a drug, as it exerts a neuroprotective effect by regulating NRSF.

## 1. Introduction

As the neuropeptide pituitary adenylate cyclase-activating polypeptide (PACAP) preferential receptor 1, PAC1-R is abundantly located in the central and peripheral nervous systems and mediates the significant anti-inflammatory [1], anti-apoptotic [2], neuroprotective [3], and neurogenetic [4] effects of PACAP. PAC1-R belongs to the class B G-protein-coupled receptors (GPCRs), which have a significantly larger N-terminal extracellular domain (PAC1-R-EC1) than other GPCR families [5]. Previous studies have demonstrated that PAC1-R mainly mediates the therapeutic effect of PACAP on Alzheimer’s disease, Parkinson’s disease, Huntington’s disease, traumatic brain injury, and stroke in in vivo and in vitro models, indicating that PAC1-R is a clinically important target for drug development for systemic degenerative diseases [6,7].

Both hydrazide and doxycycline (DOX) have been reported to bind to the PAC1-R-EC1 domain, but in different ways [8,9]. DOX, with clinical neuroprotective activity, binds PAC1-R-EC1 similarly to PACAP(30–37), while hydrazide, without clinical neuroprotective activity, binds similarly to PACAP(20–27) [9], indicating that PACAP(28–38) (GKRYKQRVKNK) plays an important role in the regulation of the activity of PACAP/PAC1-R, a finding which has also been recently recognized by other groups [10,11]. Moreover, some hydrazide and hydrazone analogs have been found to have significant neuroprotective effects, including phenyl hydrazide J147 [12], L-carnosine hydrazide [13], dantrolene-like hydrazide, and hydrazone analogues [14], etc., which are predicted to bind to the site at PAC1-R-EC1 overlapped by PACAP(28–38). These findings support the conclusion that the binding of PAC1-R-EC1 by PACAP(28–38) exerts neuroprotective activity by upregulating the activation of PAC1-R [10,11,15]. In other words, PACAP(28–38) works as a positive allosteric modulator (PAM) of PAC1-R, whereas the oligopeptides, including TAT (GRKKRRQRRRP) [15], small-molecule compounds including DOX and its derivative minocycline [9], and some analogs of hydrazide with significant neuroprotective activities mentioned above, are all considered as PAMs targeting PAC1-R. Furthermore, a novel small-molecule PAM (SPAM1) of PAC1-R, which also has a hydrazide analog-like structure, was introduced in our previous report [16]. In this report, we attempt to reveal the details of the mechanism by which PAMs targeting PAC1-R, such as SPAM1, exert neuroprotective activity by regulating the neuron-restrictive silencer factor (NRSF).

NRSF, also known as repressor element 1 (RE1), or silencing transcription factor (REST), is widely expressed during embryogenesis and binds to the neuron-restrictive silencer element (NRSE) to repress neuronal gene transcription. NRSF/REST contains two main repressor domains that restrict the expression of neuronal genes by associating with two distinct co-repressors, SIN3A and RCOR1 [17,18]. The genes reported to bind NRSF at the promoter region include the genes encoding P2RY4, NPTXR, NRXN3, NTRK3, SYN1, NEFH, SYNPHY, GRIN2a, BDNF, VIP, TH, L1CAM, etc. [19,20,21,22], which play positive roles in neuroprotection and nerve regeneration. PACAP also has three silencing elements, NRSLE1, NRSLE2, and NRSLE3, which are highly homologous to NRSE and have a similar NRSF/REST binding mode [23]. The Huntingtin (Htt) protein is mutated in Huntington’s disease (HD) [24]. It is well-known that wild-type Htt, but not the mutant form, can interact with NRSF in the cytoplasm of neurons to regulate the binding availability of REST/NRSF to its nuclear NRSE site [25]. Several studies have demonstrated the therapeutic potential of PACAP in HD. For instance, quinolinic acid (QA)-induced striatum builds an HD disease model, and pretreatment with PACAP significantly attenuates behavioral deficits and the number of damaged neurons in the striatum in QA-induced lesions [26]. PACAP treatment also rescued PAC1-R levels in R6/1 mice, promoted the expression of hippocampal BDNF, and reduced the formation of mutant Htt aggregates [27]. Our previous study demonstrated that the nuclear translocation of PAC1-R is induced by the positive allosteric regulation of PAC1-R by its PAMs, including PACAP(28–38), TAT, and DOX [15,16]. The bioinformatic analysis of ChIP-sequencing data demonstrated that the nuclear-translocated fragment of PAC1-R can bind the promoter region of PACAP and PAC1-R in some way to upregulate its expression along with the predicted transcription factors (TFs) SP1 and REST [15]. Given the therapeutic role of PACAP in HD, and SP1 as a key transcription factor for Htt expression [28], it was hypothesized that PAMs of PAC1-R, such as SPAM1, exert neuroprotective effects by interfering with the binding of NRSF through upregulating the expression of Htt by inducing nuclear translocation of the fragments of PAC1-R.

In the present work, we report that SPAM1 has a significant neuroprotective effect against oxidative stress in both H_2_O_2_-treated cell models and D-galactose (D-gal) treatment in an aging mouse model. The positive effect of SPAM1 was positively associated with increased levels of PACAP and the nuclear translocation of PAC1-R. Furthermore, the nuclear 35 kDa, but not the 15 kDa fragments, of PAC1-R can interact with SP1 to upregulate the expression of Htt, which exerts a neuroprotective effect by attenuating the binding availability of REST/NRSF with NRSE to increase the expression of NRSF/NRSE-related neuropeptides with neuroprotective activities, such as PACAP, BDNF and so on.

## 2. Results

### 2.1. Molecular Docking of SPAM1 with PAC1-R-EC1

To characterize the site recognized by PAMs targeting PAC1-R, we compared the molecular docking of all PAMs ever reported, such as PACAP(28–38), TAT, DOX, and SPAM1 [9,15,16] at PAC1-R-EC1, and PAC1-R antagonist hydrazide, which is the first small-molecule antagonist ever reported for PAC1-R [8]. As shown in Figure 1D, hydrazide binds at the junction of the extracellular N-terminus and transmembrane domain of PAC1-R, whereas SPAM1 and DOX are entirely located in the extracellular N-terminus of PAC1-R. When compared with superimposed PACAP38, SPAM1, TAT, and DOX bound to the site where PACAP(30–37) was located (indicated in the frame in Figure 1E), whereas hydrazide recognized the site where PACAP(20–27) binds. The difference in the binding sites of PAMs and non-PAMs targeting PAC1-R, indicates that binding imitating PACAP(28–38), but not PACAP(20–27), could contribute to the activity of PAMs.

When comparing the binding characteristics of SPAM1 and DOX, it was found that SPAM1 not only forms two hydrogen bonds with the carboxyl group of ASP116, but also forms hydrogen bonds with GLY114 and ASP124, respectively. In addition, SPAM1 also forms π–π water transport interactions with PHE27 and PHE115 (Figure 1A). DOX also forms two hydrogen bonds with ASP116, while having some hydrophobic interactions with CYS25, PHE27, ASN60, ASP111, GLY114, PHE115, and PHE110 (Figure 1C). The total binding interactions of SPAM1 with PAC1-R-EC1 offer a significantly higher affinity of SPAM1 than DOX for PAC1-R, which is consistent with the experimental data, which shows that SPAM1 works more effectively than DOX by targeting PAC1-R [15,16].

To further identify the key amino acid residues contributing to the efficient binding of PAMs in the PAC1-R-EC1 domain, we compared the amino acid residues contributing to the binding details of the PAMs, including SPAM1/TAT/DOX/PACAP(30–37), and the amino acid residues contributing to the binding details of non-PAM hydrazide. As shown in Table 1, three amino acid residues were shared by the four PAMs of PAC1-R, including ASP111, GLY114, and ASP116, whereas ASP116 is the only common residue that mediates the conservative hydrogen-bonding interactions of PAMs with PAC1-R-EC1. Hydrazide only binds ASP111, but not ASP116, indicating that ASP116 is the key residue mediating the effects of PAMs on PAC1-R, which is consistent with the results of our previous study [29].

### 2.2. General Procedure for the Synthesis of SPAM1

The method for preparing the small-molecule compound SPAM1 is a two-step synthesis (Figure 2A), comprising the following steps: (formula I): subjecting 2-aminobenzamide (a) and glutaric anhydride (b) to a polymerization reaction to obtain the basic compound 4-(4-oxo-3,4-dihydroquinazolin-2-yl)butyric acid (c); (formula II): 4-(4-oxo-3,4-dihydroquinazolin-2-yl)butyric acid (c), benzo[d][1,2,3]triazol-1-ol (d), 4-(aminomethyl)benzoic acid (e), and triethylamine (f) for a condensation reaction.

NMR spectra of the intermediate product (c) (Figure 2B); 1HNMR (400 MHz, DMSO): 12.16 (m, 2H), 8.08 (d, J = 5.8 Hz, 1H), 7.79–7.76 (m, 1H), 7.60 (d, J = 6.6 Hz, 1H), 7.46 (d, J = 6.0 Hz, 1H), 2.64 (t, J = 6.0 Hz, 2H), 2.32 (t, J = 5.6 Hz, 2H), 2.00–1.96 (m, 2H). NMR spectrum of SPAM1(Figure 2C). 1HNMR (400 MHz, DMSO-d6) = 12.79 (s, 1H), 12.18 (s, 1H), 8.42 (t, J = 6.0 Hz, 1H), 8.09 (d, J = 7.9 Hz, 1H), 7.90 (d, J = 7.9 Hz, 2H), 7.78 (t, J, 1H), J = 7.6 Hz, 1H), 7.61 (d, J = 8.2 Hz, 1H), 7.47 (t, J = 7.5 Hz, 1H), 7.36 (d, J = 7.9 Hz, 2H), 4.33 (d, J = 5.9 Hz, 2H), 2.64 (t, J = 7.4 Hz, 2H), 2.27 (t, J = 7.5 Hz, 2H, 2.09 = 1.95 (m, 2H).

### 2.3. SPAM1 Protects RGC-5 Cells from H_2_O_2_-Induced Neuronal Damage

As expected, the viability of RGC-5 cells decreased after H_2_O_2_ treatment, indicating that H_2_O_2_ (10 μM) was neurotoxic (Figure 3A). Annexin V-FITC/PI double staining and flow cytometry demonstrated that H_2_O_2_ significantly increased the apoptotic rate of RGC-5 cells compared to that in the control group, and that both SPAM1 and PACAP38 administration significantly reduced the apoptosis rate and SPAM1 in a concentration-dependent manner. Additionally, CCK8 results confirmed that SPAM1 (0.01–1 μM) and PACAP38 (1 nM) significantly increased cell viability, and that the cytoprotective activity of SPAM1 increased in a concentration-dependent manner, while H-SPAM1 was slightly higher than PACAP38 (Figure 3B). TUNEL assay results demonstrated that H_2_O_2_ treatment increased the number of cells stained in the TUNEL assay, and that treatment with SPAM1 and PACAP38 significantly reduced the TUNEL signal, confirming the cytoprotective effects of SPAM1 and PACAP38 against the apoptosis induced by H_2_O_2_ in the RGC-5 cells (Figure 3C).

### 2.4. SPAM1 Mitigates Oxidative Damage and Senescence of Hippocampal Neurons in D-gal-Induced Mice

To investigate the damage or senescence of hippocampal neurons in D-gal-treated mice, histopathological changes were examined by HE staining. HE staining showed significantly increased neuronal cell death in the CA3 region of hippocampal tissue from D-gal-induced aging mice. In contrast, mice subjected to SPAM1 and PACAP38 treatment exhibited a significant decrease in damaged hippocampal neurons, suggesting that SPAM1 and PACAP38 can effectively attenuate the injury of hippocampal tissue induced by D-gal in aging mice (Figure 4B).

The antioxidant effects of SPAM1 against D-gal were examined. We separately detected markers of oxidative stress in the mice brains, including SOD, T-AOC, and MDA. After 6 weeks of chronic exposure to D-gal, the SOD and T-AOC activities in the brains were significantly decreased, whereas the concentration of MDA was increased (#, *p* < 0.05, D-gal group vs. NOR, Figure 4C–E) compared to that in controls. Treatment with both SPAM1 and PACAP38 significantly promoted SOD and T-AOC activity and decreased MDA levels, whereas the antioxidant activity of SPAM1 was concentration-dependent.

### 2.5. SPAM1 Alleviated D-gal-Induced Locomotor Impairment

The locomotor function of D-gal-induced aging mice was evaluated by behavioral tests, including pole climbing, paw suspension, and open field (OF) tests. Compared to the control group, the suspension score of the D-gal group was significantly decreased (Figure 5A), T-total time (Figure 5B) and T-turn time (Figure 5C) were significantly increased (#, *p* < 0.05, D-gal vs. NOR), indicating that motor function was significantly impaired by treatment with D-gal. Treatment with both SPAM1 and PACAP38 exerted positive effects in improving the motor function impaired by D-gal, while the effect of SPAM1 displayed a concentration-dependent manner in vivo.

The SPAM1-induced effect on locomotor behavior was also evaluated with the OF test (Figure 5D) by comparing the total distance traveled (Figure 5E) and average speed (Figure 5F). D-gal significantly impaired locomotion after 42 days of treatment. There was a significant reduction in the total distance traveled and the average speed in the saline + D-gal group (#, *p* < 0.05, D-gal vs. NOR, Figure 5E,F). As expected, SPAM1 and PACAP38 treatments were able to prevent deficits in locomotor behavior in the mice, and H-SPAM1 exhibited the most effective activity, indicating that SPAM1 worked in a concentration-dependent manner.

### 2.6. SPAM1 Alleviated D-gal-Induced Memory and Cognitive Impairment

We examined the effect of SPAM1 on memory and cognitive function in aging mice using a Y-maze test and passive avoidance experiments. Figure 6A shows that the latency in the D-gal group was remarkably shorter (#, *p* < 0.05, D-gal vs. NOR) than that in the NOR group. After dosing with SPAM1 or PACAP38 for 42 days, the latency of both the three SPAM1 (L, M, and H) groups and PACAP38 group were significantly prolonged compared to that of the D-gal group. The effect of SPAM1 on working memory was investigated using a Y-maze test. D-gal significantly decreased the spontaneous alternation relative to the control group (#, *p* < 0.05, D-gal vs. NOR, Figure 6B). Moreover, treatment with SPAM1 in three doses and PACAP38 all significantly reversed the spontaneous alternation decline caused by D-gal, whereas H-SPAM1 showed the most significant effect. D-gal group had tendency of a decrease in the total arm entries, but there were no significant differences in total arm entries between the groups (Figure 6C). All above results indicate that both SPAM1 and PACAP38 could improve working memory and short-term memory in the D-gal-induced aging mouse model.

There was a significant increase in the levels of caspase-3 in the brains of mice after 6 weeks of treatment with 120 mg/kg/day D-gal (#, *p* < 0.05, D-gal vs. NOR, Figure 6D). Treatment with both SPAM1 and PACAP38 significantly decreased caspase-3 levels, and the anti-apoptotic activity of SPAM1 was concentration-dependent. The anti-apoptotic activity of SPAM1 and PACAP was consistent with their neuro-protective effects.

### 2.7. SPAM1 Prevents the Decrease in Brain PACAP Levels in D-gal-Treated Mice

To assess whether SPAM1 exerts neuroprotective activity by upregulating PACAP levels, we used both the immunohistochemical test and ELISA assay to analyze the expression level of PACAP in the hippocampus (Figure 7A,B) and used the ELISA assay to analyze the expression level of PACAP in the pituitary gland (Figure 7C).

The immunohistochemical images and corresponding statistical analysis showed that D-gal administration resulted in a significant reduction in PACAP protein expression in the hippocampus (#, *p* < 0.05, D-gal vs. NOR, Figure 7A), and that SPAM1 treatment significantly upregulated the expression of PACAP compared with the D-gal group, while H-SPAM1 had a slightly positive effect on increasing PACAP levels compared to L-SPAM1. Treatment with both SPAM1 and PACAP38 significantly attenuated the decrease in the levels of PACAP in the hippocampus induced by D-gal.

ELISA assays of PACAP in the hypothalamus and pituitary gland also demonstrated a significant reduction in PACAP levels in the D-gal group, which was ameliorated by SPAM1 in a dose-dependent manner. PACAP38 also had a significant effect on blocking the decrease in PACAP levels (Figure 7B,C).

### 2.8. SPAM1 Upregulates PACAP Expression Associated with the Nuclear Translocation of PAC1-R

The PACAP promoter-reporter vector pYr-PromDetect-PACAP containing the 2526 bp (−2500 bp to +26 bp) promoter sequence of the mouse PACAP gene, was transfected into RGC-5 cells, which were used to detect the effect of SPAM1 on PACAP promoter activity. As shown in Figure 8A, PACAP promoter activity in RGCs was significantly enhanced by SPAM1 in a concentration-dependent manner. The results of the PACAP promoter activity assay are consistent with the subsequent ELISA results for PACAP expression levels (Figure 8B).

Based on the findings of significant neuroprotective activity of SPAM1 in both in vivo and in vitro, the related working mechanism was further explored. As shown in Figure 8C, the green fluorescence signal representing PAC1-eGFP transferred and aggregated inside the nucleus of PAC1-CHO cells. In RGC-5 cells with a natural high expression of PAC1-R, incubation with SPAM1 (1 uM) for 15 min also induced the significant nuclear translocation of PAC1-R, as shown by immunofluorescence images (Figure 8D upper), and the statistical analysis (Figure 8D lower). Meanwhile, the immunofluorescence images and related statistical analysis demonstrated that SPAM1 upregulated the PAC1-R signal both in whole cells and in the nucleus (Figure 8D), indicating that SPAM1 induced a significant increase in PAC1-R levels associated with the nuclear translocation of PAC1-R.

Next, we sought to understand how the nuclear fragments of PAC1-R regulate the expression of PACAP and PAC1-R. It was hypothesized that nuclear PAC1-R might interact with some TFs, such as in our early reports on SP1, which has been reported to be the main TF that regulates the transcription of PACAP and Htt [15]. To explore the nuclear translocation of PAC1-R with respect to SP1, Co-IP was performed with antibodies targeting the C-terminus of PAC1-R or with an SP1 antibody on the nuclear extraction of RGC-5 cells treated with SPAM1 (1 uM). As shown in Figure 8E, a positive signal for SP1 was detected when it was pulled down with the PAC1-R C-terminal antibody, indicating that nuclear PAC1-R can directly interact with SP1. Meanwhile, pulling down with the SP1 antibody only detected a positive signal of 35 kDa for PAC1-R, which indicates that 35 kDa, but not 15 kDa, fragments of PAC1-R directly interact with SP1 (Figure 8D).

### 2.9. SPAM1 Exerts a Neuroprotective Effect through Neuron-Restrictive Silencer Factor

It has been reported that Htt interacts with REST/NRSF and maintains it in the cytoplasm, reducing its binding availability to the nuclear NRSE site, and ultimately allowing gene transcription [25]. We analyzed the expression levels of Htt in RGC-5 cells. The WB results demonstrated that SPAM1 (0.01–1 μM) treatment significantly increased the expression level of Htt in a concentration-dependent manner compared with the control group (Figure 9A).

Since Htt and PACAP have the same upregulation trend when treated with SPAM1, and three silencing sub-elements (NRSLE) were found in the promoter region of PACAP, we hypothesized that the upregulation of PACAP is related to the distribution of NRSF in the nucleus of RGC-5 cells. Analyses of endogenous REST/NRSF localization demonstrated that it was more abundant in the cytoplasm than in the nucleus of RGC-5 cells treated with SPAM1, whereas nuclear accumulation was observed in the control group (Figure 9B). No changes in NRSF levels were observed, suggesting that SPAM1 did not influence the expression of NRSF (Figure 9C). These results suggest that SPAM1 exerts a neuroprotective effect by promoting the expression of Htt to regulate the distribution of NRSF in the nucleus of RGC-5 cells.

We also examined whether SPAM1 upregulated other neuropeptides that have been reported to have the same NRSF binding site. We tested the expression levels of the related proteins and found that BDNF, TH, and SYN1 were upregulated by SPAM1 (Figure 9C), whereas BDNF, TH, and SYN1 had the same tendency as Htt. These results further support our hypothesis that SPAM1 exerts a neuroprotective effect through the neuron-restrictive silencer factor.

The effect of SPAM1 was compared with that of PACAP38, a neuropeptide with well-known BDNF-mediated neuroprotective effects. As expected, both SPAM1 and PACAP38 significantly upregulated Htt expression. The expression level of BDNF was significantly upregulated by PACAP38, consistent with previous reports, although the effect of PACAP38 was slightly lower than that of SPAM1. Furthermore, the expression of TH and SYN1, whose expressions were closely correlated with NRSF, were both upregulated by SPAM1 and PACAP38 (Figure 10A).

### 2.10. Neuroprotective Effect of SPAM1 Is Associated with Nuclear Translocation of PAC1-R

It has been previously confirmed that the reactive oxygen species (ROS) scavenger NAC (N-acetyl-L-cysteine) and the palmitoylation inhibitor 2-BP can inhibit the nuclear translocation of PAC1-R [30]. We now provide evidence to demonstrate that NAC (10 µM) and 2-BP (50 µM) also play an inhibitory role on the upregulation of Htt, BDNF, TH, and SYN1 induced by SPAM1 (Figure 10B). As shown in Figure 10C,D, NAC (10 µM) and 2-BP (50 µM) both can significantly affect the distribution of NRSF in the nucleus and cytoplasm, indicating that the neuroprotective effect of PAC1-R is also positively correlated with the nuclear translocation of PAC1-R. These data suggest that the neuroprotective effect of SPAM1 is positively related to the nuclear translocation of PAC1-R, and that the distribution of NRSF in the cytoplasm and nuclear regulated by Htt plays an important role in the effect of SPAM1.

## 3. Discussion

Since its isolation from the bovine pituitary gland in 1989, PACAP has been extensively studied for its neuroprotective effects with its preferred receptor PAC1-R [31]. The positive allosteric regulation of GPCRs enhances the orthotopic ligand activity without direct ligand activation [32], which may have a longer-lasting effect than positive activation. We have reported a number of small molecules and oligopeptides binding PAC1-R-EC1, which work as PAMs of PAC1-R, to exert neuroprotective effects, including DOX, minocycline, PACAP(28–38), and TAT [15,29]. Although the positive allosteric regulation of PAC1-R is recognized as a mechanism by which PAC1-R exerts a neuroprotective effect, the main mechanism is still elusive.

The results of the present study demonstrate that SPAM1 has three common binding residues, ASP111, PHE115, and ASP116, whereas ASP116 plays an important role in the hydrogen-bonding interaction with PAC1-R shared by SPAM1, TAT, PACAP(30–37), and DOX. This indicates that ASP116 is the key residue that exerts the PAM effect on PAC1-R.

Cellular data demonstrated that SPAM1 displayed effective neuroprotective activity against H_2_O_2_-induced apoptosis in RGCs, especially in a concentration-dependent manner. D-gal administration has been reported to result in accelerated aging, impaired spatial learning and memory, and excessive apoptosis of hippocampal neurons [33,34]. Similarly, in this study, we found that the cell death of neurons in hippocampal tissue and oxidative stress are increased by D-gal, which is consistent with the impairment effect produced by D-gal in brain tissues. Meanwhile, SPAM1 exerted significant effective neuroprotective activity in the mouse aging model induced by D-gal by blocking the motor and memory functional impairment induced by D-gal, promoting the activity of SOD and T-AOC, and by decreasing MDA levels and oxidative damage in hippocampal neurons. The effect of SPAM1 was demonstrated to occur in a concentration-dependent manner.

PACAP and PAC1R are expressed in the embryonic and postnatal brains of rodents [35]. During the early stages of neural development, PAC1R mRNA expression is observed in the ventricular zone throughout the neuraxis. After birth, its expression in the central nervous system is predominantly limited to the subventricular zone, olfactory bulb, hippocampus, and cerebellum [35,36]. The brain location of PACAP/PAC1-R suggests that PACAP/PAC1-R is involved not only in the locomotor activity but also in the memory activity. Our data here demonstrated for the first time that SPAM1 and PACAP38 improved both locomotor activity and the memory activity in the D-gal aging mouse model by targeting PAC1-R. Although SPAM1 and PACAP38 demonstrated similar neuroprotective activity, some difference between SPAM1 and PACAP38 was still observed, such as the PACAP38 treatment leading to a decrease in center region stay time. We consider that the reason for the difference is that the working mechanism of SPAM1 is not as same as PACAP38. PACAP38 is a full agonist of PAC1-R including both activation region and allosteric modulation region for PAC1-R; while SPAM1 works just as a positive allosteric modulator of PAC1-R imitating PACAP(28–38). SPAM1 cannot activate PAC1-R directly like PACAP, but it potentiates the activity of PAC1-R significantly.

Moreover, in this report, the impairment of locomotor activity induced by D-gal can be significantly detected while the mice was subjected to the compound movement such as suspension, turning, and short-time movement in the OF test. As for the Y-maze test, a slight but not significant decrease in the total arm entries number was induced by D-gal and may be due to the time being enough for the mice to make movements and do selection.

This study is the first to demonstrate that the effects of SPAM1 and PACAP38 are associated with PACAP upregulation. The present in vitro and in vivo data indicate that the antioxidative effects of SPAM1 and PACAP38 are positively related to the enhanced expression of PACAP associated with the peculiar domain of PACAP(28–38). The re-promotion of PACAP38 on its own expression may be the novel mechanism that helps to denote the more prolonged and more significant effect of PACAP38 than PACAP27 [10,11].

More than 30 GPCRs have been detected in cell nuclei, and it is now accepted that many GPCRs continue to signal after internalization from the plasma membrane [37]. However, the mechanisms underlying the nuclear localization of GPCRs, many of which contain putative nuclear localization signals, remain poorly understood. We first reported the nuclear translocation of PAC1-R in 2013 [38]. The downstream target genes regulated by nuclear PAC1-R are mostly involved in the process of cellular stress and are related to neuroprotection, neuronal genesis, and development, which hints that the biological effects of the nuclear translocation of PAC1-R are not limited to the increasing levels of PACAP and PAC1-R [15]. As expected, SPAM1 also significantly induced the nuclear translocation of PAC1-R in PAC1R-CHO cells with high PAC1-R expression and in mouse retinal ganglion cells (RGC-5) with natural PAC1-R expression. These data indicate that the mechanism of action of SPAM1 is the same as that of other PAMs of PAC1-R, such as PACAP(28–38) and TAT [15]. Similarly, SPAM1 can also upregulate the promoter activity and expression levels of PACAP, which further supports this hypothesis.

Our previous work has observed the fragmentation within the nucleus of PAC1-R and predicted that the nuclear PAC1-R may be enriched and functionally-associated with the transcription factor, SP1. It has also been reported that SP1 can regulate PACAP expression [15,30]. Co-IP results proved that the 35 kDa, but not the 15 kDa, fragments of PAC1-R can directly interact with SP1. Three NRSLE elements located in the PACAP promoter have attracted our attention [23]. As a key element in regulating the expression of neuronal genes, NRSE has been found in the promoter regions of many neuropeptides, including BDNF, TH, and SYN1 [19]. NRSF can specifically bind to NRSE to exert a silencing function, and this effect is regulated by wild-type Htt [25]. Given the highly potent neuroprotective effect of PAC1-R and the low levels of PAC1-R in advanced HD mice [26,27], we speculated that the nuclear translocation of PAC1-R could indirectly exert a neuroprotective effect by regulating the expression of Htt. Interestingly, SP1 is known to regulate the expression of Htt, and we detected an increase in Htt expression following treatment with SPAM1.

Moreover, cytoplasmic accumulation of NRSF occurs with the upregulation of Htt, which leads to a corresponding decrease in the nuclear availability of REST/NRSF. This results in the transcriptional increase in REST/NRSF target genes, including not only PACAP, but also other proteins such as BDNF, TH, and SYN1, almost all of which function positively in neuroprotection. As a palmitoylation disrupter interfering with the palmitoylation of the first and only single CYC25 in PAC1-R-EC1, 2-BP inhibits both the nuclear translocation of PAC1-R and its neuroprotective effects. These results confirmed that the nuclear-translocated PAC1-R fragment is responsible for the upregulation of PACAP, BDNF, and other nerve-related proteins, whose expression is regulated by REST/NRSF.

Based on the above analysis, the working mechanism of the neuroprotective effect of PAMs targeting PAC1-R, such as SPAM1, is shown in two styles: a flow chart (Figure 11A) and a cartoon diagram (Figure 11B). After the N-terminal extracellular domain of PAC1-R senses PAMs such as SPAM1 stimulation, this causes nuclear translocation under the assistance of some kind of transporter proteins, including importin/transportin/sortin/β-arrestin, depending on the C-terminal nuclear translocation sequence of PAC1-R, in some type of endosome. The C-terminus fragment of PAC1-R is cleaved and transported into the nucleus. Then, the 35 kDa nuclear PAC1-R fragment recruits SP1 to increase the expression level of Htt, which interacts with REST/NRSF and maintains it in the cytoplasm, reducing its binding availability to the nuclear NRSE site. This ultimately allows a series of nerve-related genes transcription, including PACAP.

Our findings offer not only a reasonable explanation for the amplified neuroprotective activity of small-molecule PAMs targeting PAC1-R, but also an effective working mechanism that should be used in drug development for neuroprotection through the targeting of PAC1-R.

## 4. Materials and Methods

### 4.1. Materials and Cell Lines

All of the materials for the cell culture and transfection reagents were from Invitrogen (Carlsbad, CA, USA), and reagents for the molecular biological techniques were obtained from Takara (Dalian, China) and QIAGEN (Valencia, Spain). The Chinese hamster ovary cell (CHO-K1) and mouse retinal ganglion cell (RGC-5) lines were provided by the Chinese Academy of Life Sciences (Shanghai, China). Peptide SPAM1 was synthesized by GL Biochem Ltd. (Shanghai, China) at 95% purity. Peptide purity was confirmed by reversed-phase high-performance liquid chromatography (HPLC) and characterized using matrix-assisted laser desorption/ionization time-of-flight (MALDI-TOF) mass spectrometry.

### 4.2. Cell Cultures and Transfection

CHO cells were grown in Dulbecco’s modified Eagle’s medium (DMEM) supplemented with 10% fetal bovine serum (FBS) in a humidified atmosphere of 95% air and 5% CO_2_ at 37 °C before transfection. CHO-K1 cells in the logarithmic phase were digested and seeded into six-well plates at a density of 2 × 10^5^ cells/mL. The cells were then transfected with vector constructs expressing wild-type PAC1-R tagged with eGFP at the C-terminus (PAC1R-eGFP) using Lipofectamine LTX and Opti-MEM medium (Invitrogen, Waltham, MA, USA) following the manufacturer’s instructions. For stable expression, cells transfected with plasmids were selected based on G418 (0.8–1 mg/mL) insensitivity, cloned by successive cycles of limiting dilution, and screened for the eGFP fluorescence signal. At least three cell clones expressing PAC1R-eGFP or M-PAC1R-eGFP at similar receptor levels were used in parallel in subsequent experiments.

### 4.3. Molecular Docking Utilizing Discovery Studio 2.5 (DS2.5) Software

The optimized PAC1-R 3D structure acquired from homology modeling was selected as the initial conformation for docking studies. The binding sites were defined according to the complex structure of PAC1-R-EC1 and PACAP(6–38) (PDB ID: 2JOD). Preprocessing of the PAC1-R 3D structure was implemented using DS2.5, such as hydrogenation and the application of CHARMm Forcefield (version 35b1) without CMAP backbone corrections. The 3D structures of small molecular chemicals, including SPAM1, TAT, doxycycline, and hydrazide, were sketched in DS2.5 (DS 2.5, Accelrys, San Diego, CA, USA), and the structure data files were stored following energy minimization. The docking procedure was performed using LibDock in the DS simulation software package.

### 4.4. Cell Viability Assays

Cell viability was assessed using annexin V-FITC/PI (Kaiji Biotechnology Co., Ltd., Taizhou, China), and flow cytometry was performed to detect the apoptosis rate using a BD FACSCanto II flow cytometer (Franklin Lakes, NJ, USA). RGC-5 cells were plated into a 96-well plate at a density of 1 × 10^4^ cells/well and were incubated with or without L-SPAM1 (0.01 µM), M-SPAM1 (0.1 µM), H-SPAM1 (1 µM), and PACAP38 (1 nM) for 2 h before H_2_O_2_ (10 μM) exposure. The Cell Counting Kit-8 (CCK-8) (Dojindo, Kyushu, Japan) was used to measure cell viability. After the treatment, 10 µL of CCK-8 was processed at 37 °C for 2 h for each incubation, and the absorbance was measured using an enzyme-linked immunosorbent assay (ELISA) reader at 450 nm. Cell viability was expressed as a percentage of the control group.

### 4.5. Terminal Deoxynucleotidyl Transferase dUTP Nick End Labeling (TUNEL) Assays

RGC-5 cells were incubated with or without 1 µM SPAM1 or 1 nM PACAP38 for 2 h before H_2_O_2_ (10 µM) exposure. Apoptosis was detected using the TUNEL Cell Apoptosis Detection Kit (KeyGEN Biotech, Nanjing, China), according to the manufacturer’s instructions. 4′,6-diamidino-2-phenylindole (DAPI) staining was used to determine the number of nuclei and the total cell number. All experiments were performed in at least four parallel replicates and repeated three times.

### 4.6. Animal Grouping and Treatments

Male C57/BL6 mice (6 weeks old, weights of 18–22 g) obtained from Pengyue Experimental Animal Breeding Co., Ltd. (Jinan, China) were maintained at 12:12 h light:dark cycles at 24 ± 1 °C and 55 ± 5% humidity with free access to food and water. The treatment procedure is shown in Figure 4A. Mice were randomly divided into six groups. After adapting to their new environment for 10 days, they were then subjected to two injections for 6 weeks. Two injections were used as follows: the first injection was an intraperitoneal injection with D-galactose (D-gal) (120 mg/kg/day) or saline; and the second injection was administered subcutaneously with SPAM1, PACAP38, or saline on the back of the neck of the mouse 5 min after the first injection. The six groups were: (1) normal control group (NOR): saline without D-gal; (2) D-gal group: D-gal+saline; (3) L-SMAP1 group: D-gal+SPAM1 (10 µmol/kg/day); (4) M-SMAP1 group: D-gal+SPAM1 (40 µmol/kg/day); (5) H-SMAP1 group: D-gal+SPAM1 (100 µmol/kg/day); (6) PACAP38 group: D-gal+PACAP38 (100 nmol/kg/day).

### 4.7. Tissue Preparation

After the behavioral tests were completed, the mice were euthanized and their brains were quickly removed and cleaned with a cold 0.9% saline solution. Then, the hypothalamus and pituitary gland were dissected and immediately frozen at −80 °C for subsequent analysis. Brain tissue samples were thawed and homogenized in a cold 0.9% saline solution.

### 4.8. Hematoxylin-Eosin (HE) Staining

Sixty 3-μm sections were obtained from each paraffin block using a microtome and stained with HE. The samples were then immersed in xylene and alcohol, stained with hematoxylin for 5 min, stained with eosin for 3 min, and re-immersed in alcohol and xylene. The slides were mounted using a synthetic resin (Entellan, Merck, Darmstadt, Germany). Slices were made from numbers 1 (most superficial) to 60 (deepest).

### 4.9. Antioxidative Activity Evaluation

After excision of the hypothalamus and pituitary gland, brain tissue was homogenized in ice-cold saline with a tissue homogenizer. The whole-brain tissue homogenate, without the hippocampus, was centrifuged at 3000 rpm for 10 min, and the supernatant was subjected to measurement of superoxide dismutase (SOD) activity, malondialdehyde (MDA) level, and total antioxidant capacity (T-AOC) using SOD, MDA, and T-AOC kits (Jiancheng Biotechnology, Nanjing, China), according to the manufacturer’s instructions. All antioxidative data was normalized to the protein concentration of the supernatant and quantified using a BCA protein assay kit. All experiments were performed in at least four parallel replicates and repeated three times.

### 4.10. Pole-Climbing Tests

The apparatus for the pole-climbing test was a 50 cm-high gauze-taped (diameter, 1 cm) pole. On the 52nd day, 2 h after the last i.p. injection, a pole-climbing test was conducted. The time spent by each mouse to complete the turning orientation (T-turn) and the time spent to complete both the turning orientation and descending pole (T-total) was counted. Meanwhile, the tests were recorded via an overhead digital video camera, and the results were plotted as the average T-turn time and average T-total time from three repeated tests.

### 4.11. Paw-Suspension Tests

The apparatus for the paw-suspension test was a horizontal metal wire with a diameter of l mm placed at a height of 30 cm above the ground. Each mouse was graded based on its performance. The mice able to grasp the metal wire with two front paws received a score of 3, those able to grasp the metal wire with one front paw received a score of 2, those who struggled but could not grasp the metal wire with either front paw received a score of 1, and those who dropped directly without any grasp received a score of 0. The results were plotted as the average scores from three repeated tests.

### 4.12. Open Field Tests

The OF test was conducted using a square box made of grey plexiglass plastic (30 cm × 30 cm × 30 cm). Individually, the mice were placed in the center of the OF and allowed to freely explore for 5 min while being recorded. The OF was cleaned thoroughly between each mouse to eliminate excretions and odor cues using 70% ethanol and allowed to air-dry. The FIJI/ImageJ plugin mouse behavioral analysis toolbox (MouBeAT) software ver. The 1.00 was used to analyze the videos [39]. Locomotor activity was determined by the total distance traveled and the average speed of the mice, as calculated by MouBeAt.

### 4.13. Passive Avoidance

All mice were subjected to a passive avoidance test, according to previous studies [40]. The shuttle box included two identical compartments (360 mm × 300 mm × 130 mm) on the left and right sides, with the right compartment being a dark box. The two compartments were connected by an arched door, and the fence floor of the black box was connected to a stimulator to provide a continuous electric shock (0.3 mA). At the beginning of the training, the mice were placed in a bright box and allowed to adapt to the environment for 3 min. The mice were subjected to an electric shock once they entered the dark box. The duration of the experiment for each mouse was 5 min. The mice were tested 24 h after training. The time when the mice entered the dark box was recorded as the latency and the number of times the mice entered the dark box within 5 min was recorded. The latency of mice that did not enter the dark box within 5 min was calculated at 300 s.

### 4.14. Y-Maze Tests

The Y-maze is a hippocampal-dependent spatial working memory task that requires mice to use external maze cues to navigate through identical internal arms. The Y-maze test was performed after NOR, according to a previously described method [41]. The Y-maze contained three dark polyvinyl plastic arms with a 120° angle between all arms, which were 30 cm long, 5 cm wide, with 12 cm- high walls. The mice were initially placed at the end of one arm and allowed to explore freely in the Y-maze. The number of arm entries and sequence of arm visits were recorded manually for each mouse for 8 min. The mouse was then removed from the Y-maze to its house, and the Y-maze was cleaned with 70% ethanol to remove the odor until dry. The mouse consecutively entering three different arms was defined as an actual alteration, that is, BCA, ABC, or CAB, but not ABB. The percentage of alteration was calculated as the number of actual alterations/total number of entries × 100.

### 4.15. Caspase-3 Activity Analysis

Caspase-3 activity was determined using a caspase-3 assay kit (Beyotime Institute of Biotechnology, Haimen, China), according to the manufacturer’s instructions. Briefly, cells were lysed and centrifuged to obtain supernatants, which were then mixed with a buffer containing the substrate peptides for caspase-3 attached to p-nitroanilide and incubated for 2 h at 37 °C. OD was determined by measuring the absorbance at 405 nm (OD405) using a microplate reader. Caspase-3 activity was normalized to the protein concentration in the supernatant and quantified using a BCA protein assay kit. All experiments were performed in at least four parallel replicates and repeated three times.

### 4.16. Immunohistochemistry

Hippocampal tissue samples were fixed in 4% paraformaldehyde at 4 °C for 48 h before dehydration and embedding in paraffin. Next, 5 µm-thick coronal sections were cut using a microtome and mounted on glass slides. Sections were deparaffinized in xylene and rehydrated with decreasing concentrations of ethanol. Polyclonal anti-PACAP38 antibody (ab216627; Abcam, Cambridge, UK) was used to detect PACAP. An ABC Elite kit (Vector Laboratories, Newark, CA, USA) was used for immunohistochemical staining. PACAP-positive cells were visualized with 3,3V-diaminoben-zidine tetrahydrochloride, and the tissue sections were counterstained with hematoxylin. ImageJ was used for semi-quantification of DAB-positive cells normalized to nuclei (hematoxylin staining).

### 4.17. Fluorescence Confocal Microscopy

Cellular trafficking of PAC1R-eGFP was evaluated after the cells were treated with or without SPAM1 (1 µM) for 15 min by visualizing the fluorescence of eGFP using the appropriate spectral settings (excitation, 488 nm argon laser; emission, 545 nm filter; pinhole diameter, 2.3 airy units) of the confocal microscope (LSM 510 META; Zeiss, Thornwood, NY, USA) equipped with a Plan-Apochromat636/1.4 numerical aperture oil objective. Fluorescence confocal images were taken and analyzed using ImageJ 3.0. Nuclear translocation efficiency of PAC1-R was defined as fluorescence intensity in the nucleus/fluorescence intensity of the whole cell. The experiments were performed in parallel with at least three replicates and were repeated three times.

### 4.18. Immunofluorescence

Cellular trafficking of PAC1-R in RGC-5 cells was evaluated by immunofluorescence confocal imaging. RGC-5 cells grown in F12 medium with 10% FBS at the logarithmic phase (the confluence of cells reached 70% to 80%) were inoculated into confocal petri dishes at a density of 2 × 10^4^ cells/well. Experimental groups were treated with SPAM1 (1 µM) for 15 min, washed twice with PBS, and incubated with 4% paraformaldehyde and 0.2% Triton X-100 for 5 min at 37 °C. Following blocking, cells were incubated with rabbit polyclonal antibody raised against human PAC1-R/ADCYAP1R1 506–525 aa (LSKSSSQIRMSGLPADNLAT) (ab28670, Abcam, Cambridge, UK), which is the same as the 428–447 aa of the mouse PAC1-R isoform (normal/hop) splicing variant used in this research, for 1 h at room temperature, washed twice with PBS, and incubated for 1 h with Alexa Fluor^®^ 647-conjugated anti-rabbit antiserum (ab150075, Abcam, Cambridge, UK). Images were taken using a confocal microscope (LSM 510 META; Zeiss, Thornwood, NY, USA) with excitation at 647 nm and emission at 674 nm and subjected to Image J 3.0 analysis to determine the location and the density of fluorescence-presenting PAC1-R. The nuclear translocation efficiency of PAC1-R was defined as fluorescence intensity in the nucleus/fluorescence intensity of the whole cell. The data were plotted as the fold-change of the nuclear translocation efficiency of PAC1-R in cells without any treatment. With respect to the assessment of the expression level of PAC1-R, based on consistent and standard operational processes, the PAC1-R expression level of each cell was indicated by the fluorescence intensity of the cell, and the PAC1-R expression level of each treatment was represented by the average fluorescence intensity from ten cells of each treatment. The experiments were performed in parallel with at least three replicates and were repeated three times.

### 4.19. PACAP Promoter Activity Assay Using Recombinant Vector pYr-PromDetect-PACAP

According to the sequence from −2500 to +26 bp of the transcription initiation site of the mouse PACAP gene, we synthesized the 2526 bp promoter sequence of PACAP and cloned it into a pUC57-Simple plasmid for sequencing characterization. The PACAP promoter sequence was cleaved from pUC57-Simple-PACAP by BglII and XbaI, and ligated into the promoter-reporter vector pYr-PromDetect (Yingrun Biological Company, Changsha, China) cut by BglII and NheI to construct the recombinant vector pYr-PromDetect-PACAP. The characterization of pYr-PromDetect-PACAP was conducted by restriction cleavage with BamHI, which theoretically produces two bands.

For the assay of PACAP promoter activity, RGC-5 cells in the logarithmic phase were seeded into 24-well plates at a density of 5 × 10^3^ cells/well and incubated to 70–80% confluency. The recombinant vector pYr-PromDetect-PACAP (500 ng/well), or the blank vector (pYr-PromDetect) (500 ng/well) used as a control, was transfected into RGC-5 cells using Lipofectamine LTX & PLUS Reagent (Invitrogen, Waltham, MA, USA). After transfection, cells were incubated in F12 medium containing 10% FBS for 48 h. Then, the culture medium was removed and serum-free F12 culture was used, and the experimental groups were treated with a concentration gradient of SPAM1 (0.01–1 µM) for 15 min. Fetal bovine serum-free basal medium was used for the control group, and the cells were washed twice with PBS after incubation. Then, the cells were treated according to the manufacturer’s instructions for the Dual-Luciferase^®^ Reporter Assay System (Promega, Madison, WI, USA), and the activities of the two luciferases were assayed using the Victor3142 multi-label counter (PerkinElmer, Waltham, MA, USA). Promoter activity was first expressed as the ratio of Renilla luciferase activity (promoter-reporter gene) to firefly luciferase activity (reference gene), and was then presented as the fold-change of the data from the cells transfected with the blank vector pYr-PromDetect. The experiments were performed in parallel with at least three replicates and were repeated three times.

### 4.20. PACAP Levels Determined by ELISA

RGC-5 cells were plated into a 12-well plate at a density of 1 × 10^5^ cells/well, and the experimental groups were treated with a concentration gradient of SPAM1 (0.01–1 µM) and incubated for 30 min. Fetal bovine serum-free basal medium was used for the control group, and the cells were washed twice with PBS after incubation. PACAP concentration in RGC-5 cells was assayed using a PACAP ELISA kit (Lvyuan Bird Biotechnology, Beijing, China). PACAP levels were further normalized to the protein concentration quantified using the BCA protein assay kit (Thermo Fisher, Waltham, MA, USA). All experiments were performed in at least four parallel replicates and repeated three times.

### 4.21. Co-Immunoprecipitation (Co-IP)

The nuclear fraction of RGC-5 cells was prepared using a KeyGEN Nuclear and Cytoplasmic Protein Extraction Kit (KeyGEN, Shanghai, China). Two hundred microliters of the supernatant was added to 20 μL Protein A/G (Abmart, Shanghai, China) and incubated with the anti-PAC1-R/ADCYAP1R1 506–525 aa antibody (Abcam, Cambridge, UK) or anti-SP1 antibody (Affinity, Jiangsu, China) overnight at 4 °C. The next day, the immunoprecipitants were washed five times and boiled in 1× SDS loading buffer. Western blotting was performed after SDS-PAGE. The experiments were performed in parallel with at least three replicates and were repeated three times.

### 4.22. Western Blotting Assays

Logarithmic phase RGC-5 cells were seeded into six-well plates with 2 × 10^5^ cells/well and incubated with F12 medium containing 10% FBS at 37 °C, 5% CO_2_, and 80% confluence. The experimental groups were treated with SPAM1 (0.01–1 µM) and PACAP38 (1 µM) with or without N-acetyl-L-cysteine (NAC) (10 nM) or 2-bromopalmitate (2-BP) (50 µM) and incubated for 30 min. The control group was treated with fetal bovine serum-free basal medium. Total protein was extracted from the cells using RIPA buffer (50 mM Tris-HCl (pH 7.4), 150 mM NaCl, 20 mM EDTA, 1% Triton X-100, 1% sodium deoxycholate, 1% SDS, and protease inhibitors (Beyotime, Shanghai, China) on ice for 30 min. The plasma fraction without the nuclear fraction and the nuclear fraction of RGC-5 cells were prepared using the KeyGEN Nuclear and Cytoplasmic Protein Extraction Kit (KeyGEN, Shanghai, China) and analyzed by SDS-PAGE. Membranes were incubated with rabbit polyclonal antibodies, as follows: anti-PAC1-R antibody (Abcam, Cambridge, UK), anti-SP1 antibody (Affinity, Jiangsu, China), anti-Htt antibody (Abclonal, Wuhan, China), anti-NRSF antibody (HUABIO, Zhejiang, China), anti-SYN1 antibody (SAB, Greenbelt, Maryland, USA), anti-tyrosine hydroxylase antibody (SAB), anti-BDNF antibody (HUABIO), anti-β-actin antibody (ab227387, Abcam), and anti-histone H1 antibody (SAB), followed by incubation with HRP-conjugated secondary antibody (ThermoFisher, Waltham, MA, USA). Protein bands were visualized using an enhanced chemiluminescence (ECL) kit (Beyotime Biotechnology).

### 4.23. Statistical Analyses

Statistical analyses were performed using GraphPad Prism 5 (Graph Pad Software, San Diego, CA, USA; http://www.graphpad.comgraphpad.com (accessed on 1 January 2020). All data are presented as mean ± standard error of the mean (SEM). One-way analysis of variance (ANOVA) was used to evaluate the differences among the groups. Differences were considered statistically significant at *p* < 0.05.

## 5. Conclusions

In summary, a new small molecule positive allosteric modulator of PAC1-R, named SPAM1, shows the effective neuroprotective activities against oxidative stress both in an H_2_O_2_-treated cell model and in D-gal-treated aging mouse models. The positive effect of SPAM1 was accompanied by an increased level of PACAP and the nuclear translocation of PAC1-R. PAC1-R in the nucleus was fragmented and the nuclear 35 kDa fragment of PAC1-R can interact with SP1 to upregulate the expression of Htt. The increased level of Htt in the cytoplasm inhibits the nuclear translocation of NRSF and attenuates the interaction between NRSF and NRSE, leading to the upregulation of a series of nerve-related proteins regulated by NRSF/NRSE, including PACAP, BDNF, TH and SYN1. In short, this study describes a novel mechanism by which PAMs of PAC1-R, such as SPAM1, exert neuroprotective effects by inducing the nuclear translocation of PAC1-R and increasing the expression of nerve-related genes by regulating NRSF distribution.

## Figures and Tables

**Figure 1 ijms-23-15996-f001:**
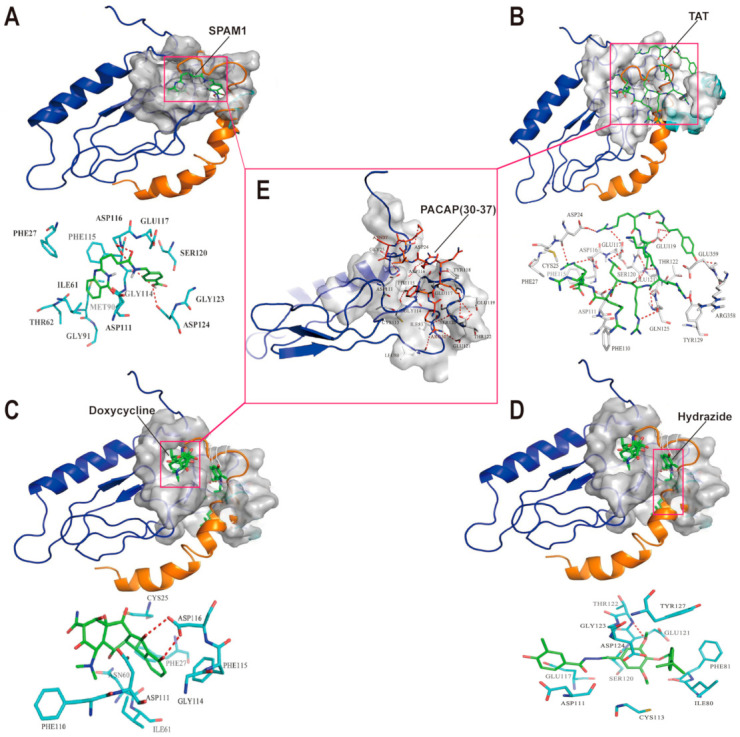
The predicted molecular docking of SPAM1, TAT, DOX, and Hydrazide on PAC1-R and their binding sites compared with PACAP(30–37). (**A**) The upper showed the predicted binding site of SPAM1 on PAC1-R represented by the surface model and the lower showed the details of the predicted binding mode; (**B**) the upper showed the predicted binding site of TAT on PAC1-R represented by the surface model and the lower showed the details of the predicted binding mode; (**C**) the upper showed the predicted binding site of DOX on PAC1-R represented by the surface model and the lower showed the details of the predicted binding mode; (**D**) the upper showed the predicted binding site of hydrazide on PAC1-R represented by the surface model and the lower showed the details of the predicted binding mode; (**E**) PAC1-R with PACAP(30–37) superimposed according to the resolved complex 3D structure of PACAP38 with the extracellular domain of PAC1-R (PDB ID: 2JOD). All the binding sites are represented by the surface model. PAC1-R in deep blue and PACAP38 in yellow are shown in cartoon style, and the binding small molecules are shown as sticks. All the contact residues are shown and labeled by type and number, and the red dotted line illustrated the hydrogen bond interaction. The middle frame indicated that SPAM1, TAT, and DOX shared a similar binding site recognized by PACAP(30–37).

**Figure 2 ijms-23-15996-f002:**
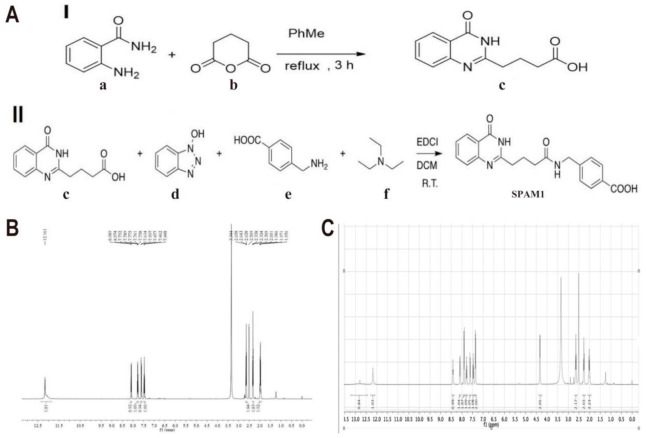
Synthesis process and NMR spectrum of the small molecule SPAM1. (**A**) Synthesis process of the small molecule SPAM1 (a two-step method). a: 2-aminobenzamide; b: glutaric anhydride; c: 4-(4-oxo-3,4-dihydroquinazolin-2-yl) butanoic acid; d: benzo[d][1,2,3]triazol-1-ol; e: 4-(aminomethyl)benzoic acid; f: triethylamine; (**B**) NMR spectrum of the intermediate product c; 1HNMR (400 MHz, DMSO): 12.16 (m, 2H), 8.08 (d, J = 5.8 Hz, 1H), 7.79–7.76 (m, 1H), 7.60 (d, J = 6.6 Hz, 1H), 7.46 (d, J = 6.0 Hz, 1H), 2.64 (t, J = 6.0 Hz, 2H), 2.32 (t, J = 5.6 Hz, 2H), 2.00–1.96 (m, 2H). (**C**) NMR spectrum of the SPAM1. 1HNMR (400 MHz, DMSO-d6) = 12.79 (s, 1H), 12.18 (s, 1H), 8.42 (t, J = 6.0 Hz, 1H), 8.09 (d, J = 7.9 Hz, 1H), 7.90 (d, J = 7.9 Hz, 2H), 7.78 (t, J, 1H), J = 7.6 Hz, 1H), 7.61 (d, J = 8.2 Hz, 1H), 7.47 (t, J = 7.5 Hz, 1H), 7.36 (d, J = 7.9 Hz, 2H), 4.33 (d, J = 5.9 Hz, 2H), 2.64 (t, J = 7.4 Hz, 2H), 2.27 (t, J = 7.5 Hz, 2H, 2.09 = 1.95 (m, 2H).

**Figure 3 ijms-23-15996-f003:**
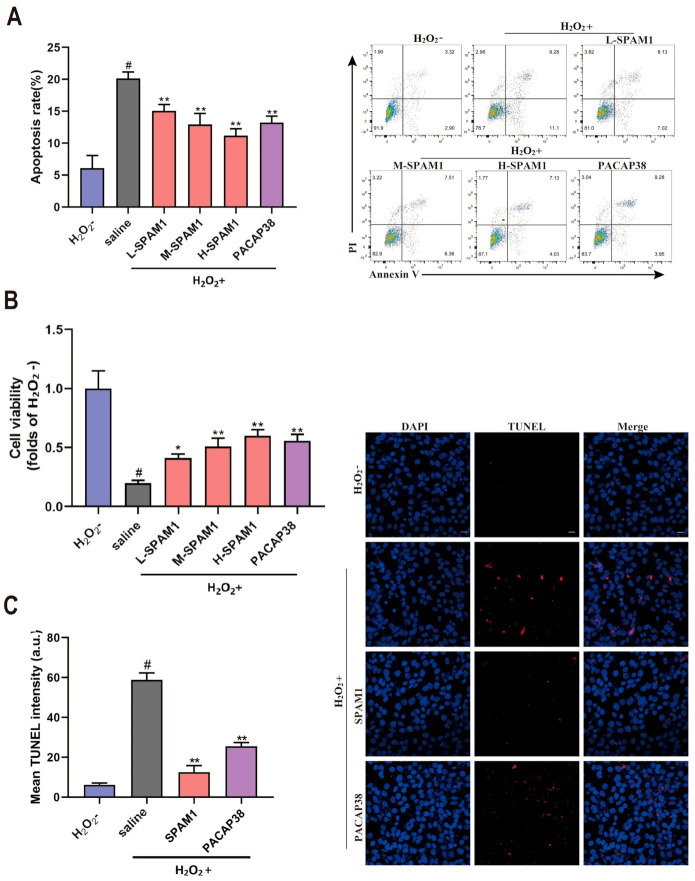
The cytoprotective activity of SPAM1. (**A**) The data analysis (**left**) of Annexin V-FITC/PI double staining and flow cytometry (**right**) showed that SPAM1(0.01–1 μM) and PACAP38(1 nM) significantly inhibited apoptosis induced by H_2_O_2_; (**B**) CCK8 results confirmed that SPAM1(0.01–1 μM) and PACAP38(1 nM) significantly increased cell viability that decreased by H_2_O_2_; (**C**) effect of SPAM1 and PACAP38 on apoptosis determined by TUNEL assay and corresponding analysis (**left**) of the TUNEL images (**right**) in RGC-5 cells induced by H_2_O_2_. It was shown that SPAM1(0.01–1 μM) and PACAP38(1 nM) has significantly cytoprotective activity against the apoptosis induced byH_2_O_2_. #, *p* < 0.05, H_2_O_2_+ group vs. H_2_O_2_− group; *, *p* < 0.05, vs. H_2_O_2_+ group; **, *p* < 0.01, vs. H_2_O_2_+ group. Data are presented as the mean ± SEM of three experiments (Scale bars: 20 μm).

**Figure 4 ijms-23-15996-f004:**
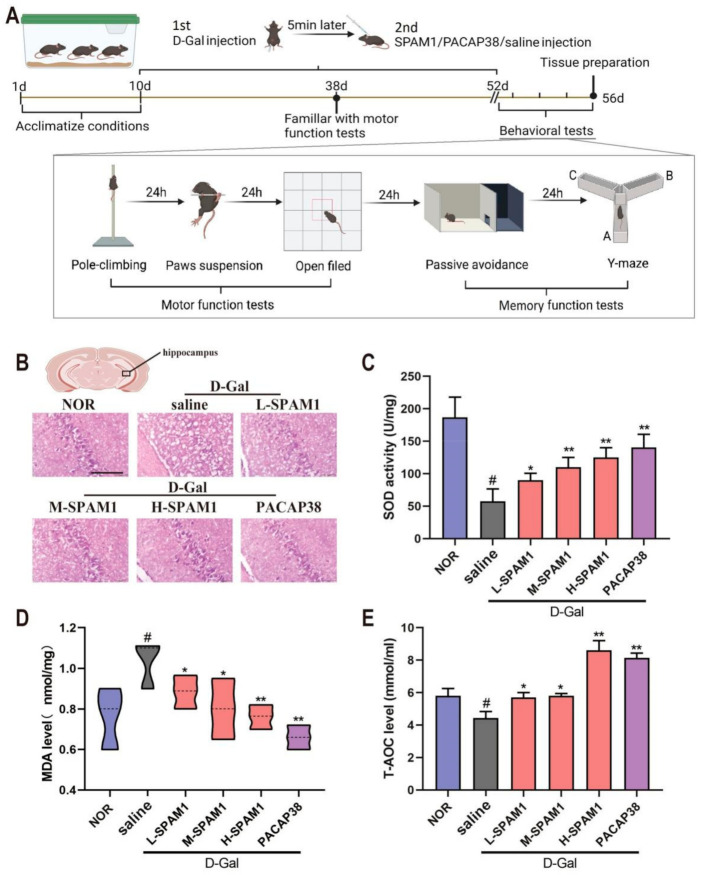
Effect of SPAM1 on antioxidative status of D-gal aging mouse brain. (**A**) The mouse D-gal aging model was firstly subjected to the treatment with L-SMAP1 (10 µmol/kg/day), M-SMAP1 (40 µmol/kg/day, H-SMAP1 (100 µmol/kg/day) and PACAP3 (100 nmol/kg/day), respectively, and then submitted to motor function and memory function test procedure; (**B**) HE staining showed that SPAM1 (L, M and H) and PACAP38 treatment exhibited significant increase in HE signal compared to the damaged hippocampal neurons induce by D-gal. The antioxidant index assays confirmed that the SOD activity decreased (**C**), MDA level increased (**D**), and T-AOC decreased (**E**) significantly in the mouse brain treated with D-Gal. SPAM1 (L, M and H) and PACAP38 significantly inhibited the negative effect induced by D-Gal. H-SPAM1 showed the most effective antioxidative activity, indicating that SPAM1 work in concentration-dependent manner. #, *p* < 0.05, D-Gal group vs. control group (NOR); *, *p* < 0.05, vs. D-Gal group; **, *p* < 0.01, vs. D-Gal group. Data are presented as the mean ± SEM, *n* ≥ 6 (Scale bars: 100 μm).

**Figure 5 ijms-23-15996-f005:**
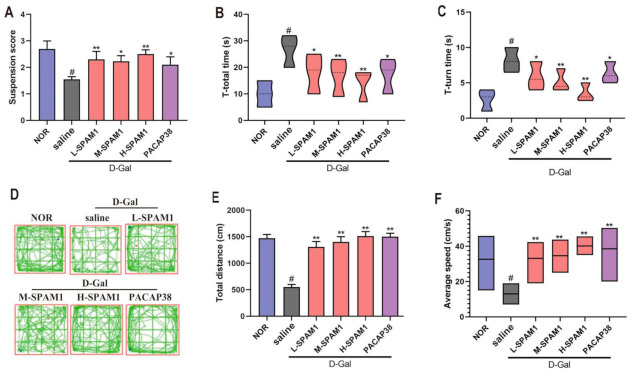
Effect of SPAM1 and PACAP38 on D-Gal-induced motor function impairment. The mouse D-Gal aging model were subjected to the treatment with L-SMAP1 (10 µmol/kg/day), M-SMAP1 (40 µmol/kg/day, H-SMAP1 (100 µmol/kg/day) and PACAP38 (100 nmol/kg/day), respectively. (**A**) Paw suspension score showed that L-SPAM1, M-SPAM1, H-SPAM1, and PACAP38 all significantly inhibited the decrease in suspension score induced by D-gal. The average time to complete both T-turn and descending (T-total) (**B**) and average time just to complete T-turn (**C**) also showed the protective effect of SPAM1 and PACAP38 against D-Gal-induced motor impairment. (**D**) Representative track maps generated from MouBeAt Software illustrated the locomotor pattern of mice during 5 min spent in the OF; locomotor activity was assessed by comparing the total distance traveled (**E**) and average speed (**F**) of mice. It was shown that SPAM1 (L, M and H) and PACAP38 significantly protected mice from the impairment of D-gal on motor functions. H-SPAM1 exhibited the most efficient protective activity, indicating SPAM1 work in concentration-dependent manner. #, *p* < 0.05, D-Gal group vs. control group (NOR); *, *p* < 0.05, vs. D-Gal group; **, *p* < 0.01, vs. D-Gal group. Data are presented as the mean ± SEM, *n* = 10.

**Figure 6 ijms-23-15996-f006:**
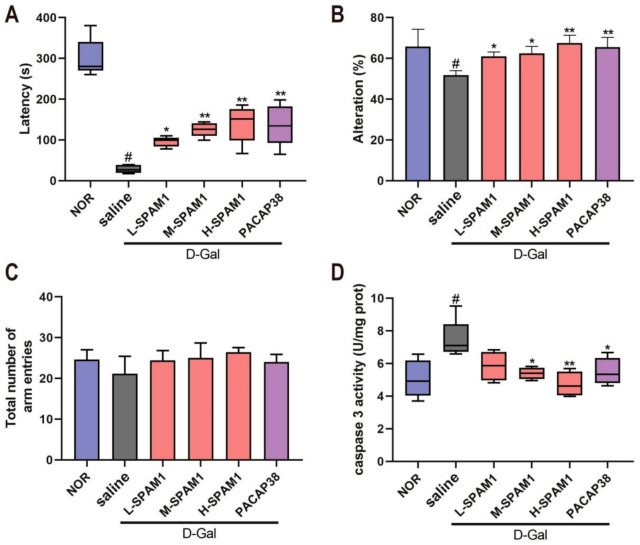
Effect of SPAM1 and PACAP38 on D-Gal-induced memory and cognitive impairment. The mouse D-Gal aging model treated with L-SMAP1 (10 µmol/kg/day), M-SMAP1 (40 µmol/kg/day, H-SMAP1 (100 µmol/kg/day) and PACAP3 (100 nmol/kg/day), respectively, was submitted to passive avoidance experiment and Y-maze test. (**A**) The latency in passive avoidance experiments demonstrated that SPAM1 (L, M and H) and PACAP38 ameliorated the memory impairment by D-gal significantly. Spontaneous alternation (**B**) and the number of total arm entries (**C**) from Y-maze test are shown along with the caspase 3 activity in the mice brain (**D**). #, *p* < 0.05, D-Gal group vs. control group (NOR); *, *p* < 0.05, vs. D-Gal group; **, *p* < 0.01, vs. D-Gal group. Data are presented as the mean ± SEM, *n* = 10.

**Figure 7 ijms-23-15996-f007:**
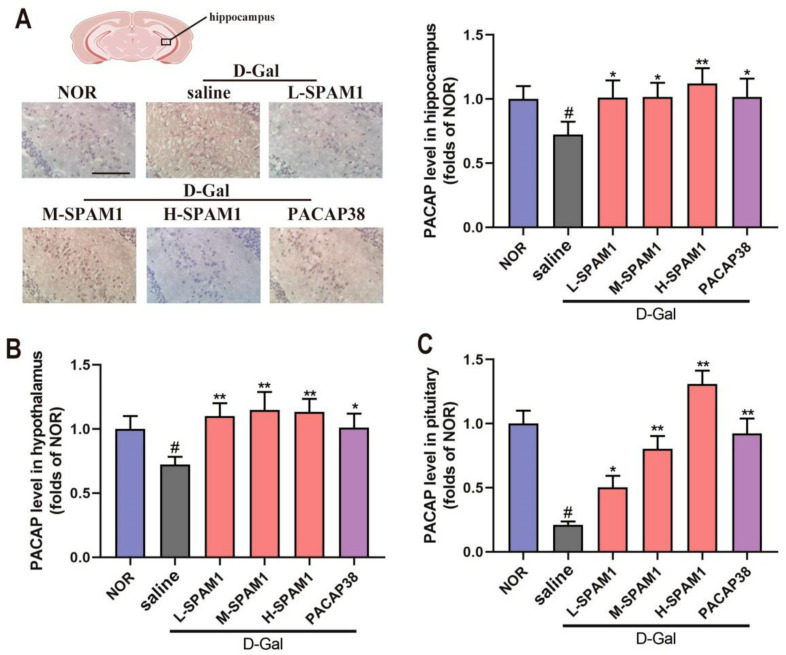
Effect of SPAM1 and PACAP38 on the levels of PACAP in hippocampus and pituitary of D-Gal-treated mice treated with L-SMAP1 (10 µmol/kg/day), M-SMAP1 (40 µmol/kg/day, H-SMAP1 (100 µmol/kg/day), and PACAP3 (100 nmol/kg/day), respectively. (**A**) The immunohistochemical image (**left**) and corresponding quantitative measurement of staining intensity of PACAP (**right**) in hypothalamus; (**B**) the PACAP level in hippocampus tested by Elisa assay; (**C**) the PACAP level in pituitary tested by Elisa assay. As shown, SPAM1 and PACAP38 treatment significantly increase the levels of PACAP in both hippocampus and pituitary compared with the D-Gal group, while SPAM1 worked in dose-dependent manner. #, *p* < 0.05, D-Gal group vs. control group (NOR); *, *p* < 0.05, vs. D-Gal group; **, *p* < 0.01, vs. D-Gal group. Data are presented as the mean ± SE, *n* ≥ 6 (Scale bars: 100 μm).

**Figure 8 ijms-23-15996-f008:**
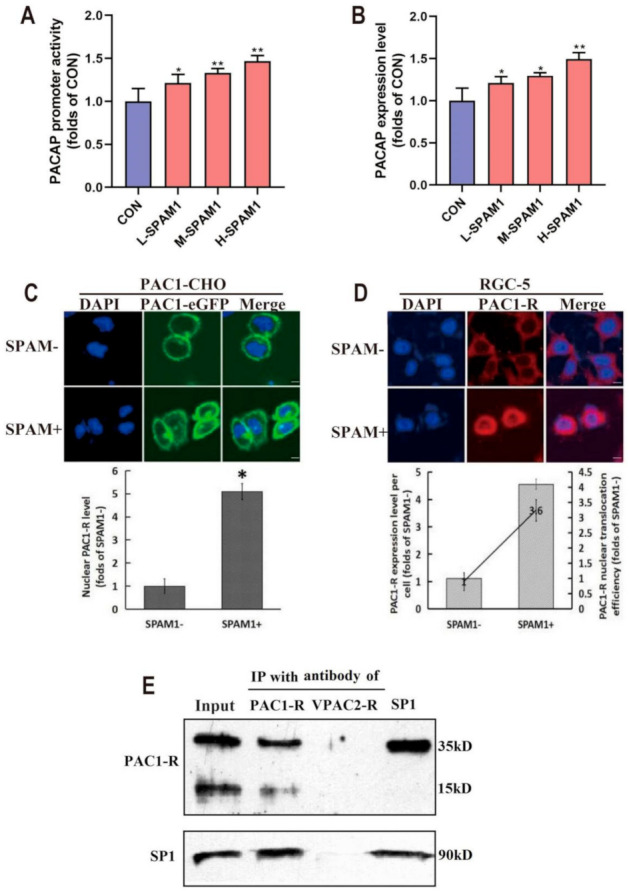
Effect of SPAM1 on the nuclear translocation of PAC1-R. (**A**) The dual-luciferases reporter assay result demonstrated that the promoter activity of PAC1-R is significantly enhanced by SPAM1 (0.01–1 µM). (**B**) The ELISA result showed that SPAM1 (0.01–1 µM) significantly increase the expression level of PACAP in RGC-5 cells. (**C**) The effect of SPAM1 (1 uM) on the nuclear translocation of PAC1R-eGFP detected in PAC1R-CHO cells by fluorescence images (**up**) and corresponding statistical analysis (**down**). Shown was SPAM1 (1 uM) inducing significant traffic and aggregation of PAC1R-eGFP fluorescence signal (green) into the nucleus. (**D**) Immunofluorescence images with the antibody targeting the C-terminus of PAC1-R (**up**) and the related statistical analysis of PAC1-R signals (**down**) in the RGC-5 cells treated with SPAM1 (1 uM). In the statistical analysis (**down**), the line with the left axis represents the PAC1-R expression level per cell and the column with the right axis indicated PAC1-R nuclear translocation efficiency. It was demonstrated that SPAM1 (1 uM) significantly up-regulated the expression of PAC1-R associated with the nuclear translocation of PAC1-R. (**E**) The Co-IP assay result detected the positive signal of SP1 in pulling down with the PAC1-R C-terminal antibody; meanwhile, no PAC1-R signal or SP1 signal was detected in pulling down with VPAC2-R antibody, indicating that nuclear PAC1-R can directly interact with SP1. The Re-IP assay result only detected a positive signal of 35 kDa of PAC1-R in pulling down with SP1 antibody, which indicate 35 kDa but not 15 kDa fragments of PAC1-R directly interact with SP1. *, *p* < 0.05, vs. CON; **, *p* < 0.01 vs. CON. Data are presented as the mean ± SEM of three experiments (Scale bars, 5 um).

**Figure 9 ijms-23-15996-f009:**
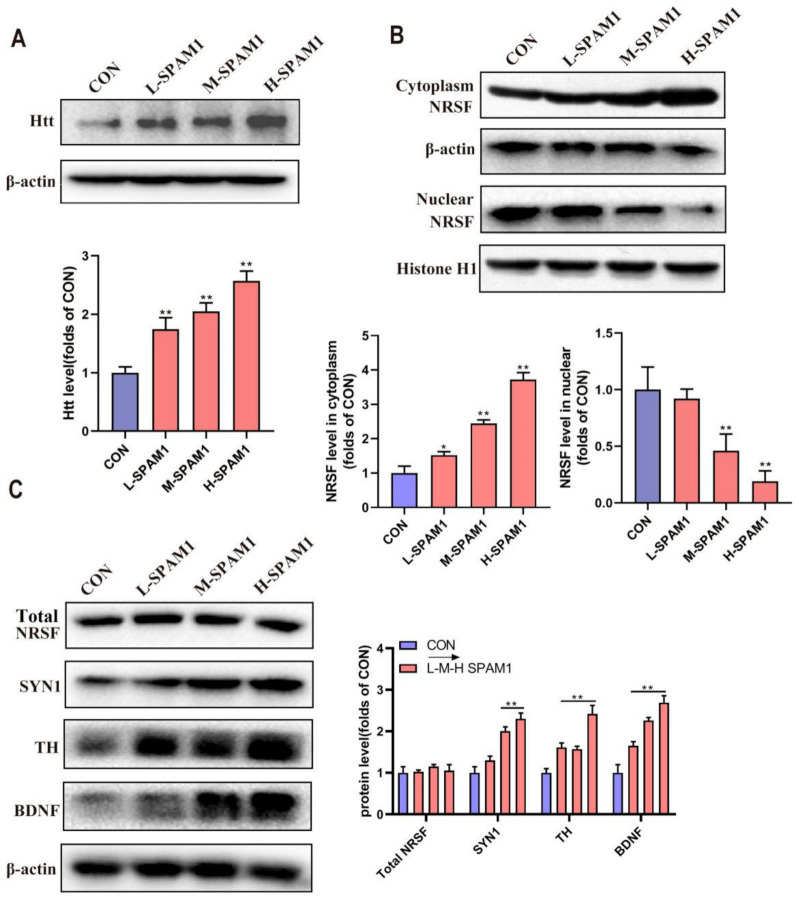
The effect of SPAM1 on Htt level, on the distribution of NRSF between cytoplasma and nuclear and on the expression level of neuropeptides including SYN1, TH, and BDNF in RGC-5 cells. (**A**) The WB images of Htt in whole RGC-5 cells incubated with SPAM1 in concentration from 0.01 uM to 1 uM (**up**) and the corresponding statistical analysis (**down**) showed that level of Htt increased following the increase in the SPAM1 concentration. (**B**) The WB images of NRSF in cytoplasma and nuclear (**up**) and the corresponding statistical analysis (**down**) showed cytoplasm NRSF level increased following the treatment with SPAM1 and nuclear NRSF level have an opposite trend in SPAM1 concentration-dependent manner. (**C**) The Western blot results (**left**) and the subsequent quantitative analysis (**right**) demonstrated that SYN1, TH, and BDNF were significantly up-regulated by SPAM1 following the increase in SPAM1 concentration. No change in the whole cellular NRSF (total NRSF) levels was observed, indicating that SPAM1 does not interfere with the expression of NRSF, but influences the distribution of NRSF between cytoplasma and nuclear. *, *p* < 0.05, vs. CON; **, *p* < 0.01, vs. CON. Data are presented as the mean ± SEM of three experiments.

**Figure 10 ijms-23-15996-f010:**
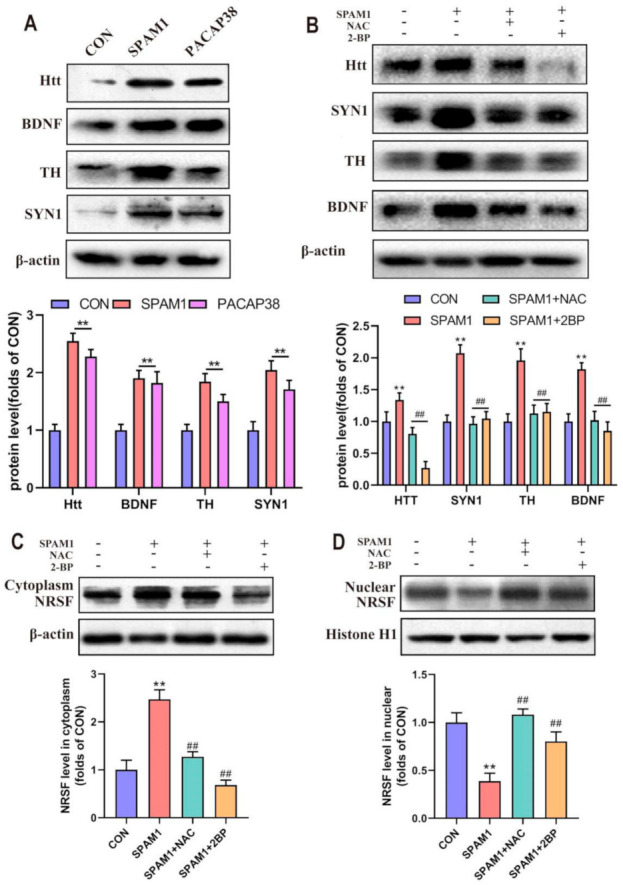
Neuroprotective effect of SPAM1 are associated with the nuclear translocation of PAC1-R. (**A**) The Western blot results (**upper**) and the subsequent quantitative analysis (**down**) showed that both SPAM1 and PACAP38 significantly up-regulated the expression of Htt and the proteins whose gene transcriptions are involved with NRSF, such as BDNF, TH, and SYN1. (**B**) The Western blot results (**upper**) and the subsequent quantitative analysis (**down**) showed that the Htt, BDNF, TH, and SYN1 expression level increased by SPAM1 treatment was significantly inhibited by NAC and 2-BP; (**C**,**D**) the retention of NRSF in cytoplasm and decrease in NRSF in nuclear induced by SPAM1 were both significantly inhibited by 2-BP and NAC. **, *p* < 0.01, vs. CON; ##, *p* < 0.01, vs. SPAM1. Data are presented as the mean ± SEM of three experiments.

**Figure 11 ijms-23-15996-f011:**
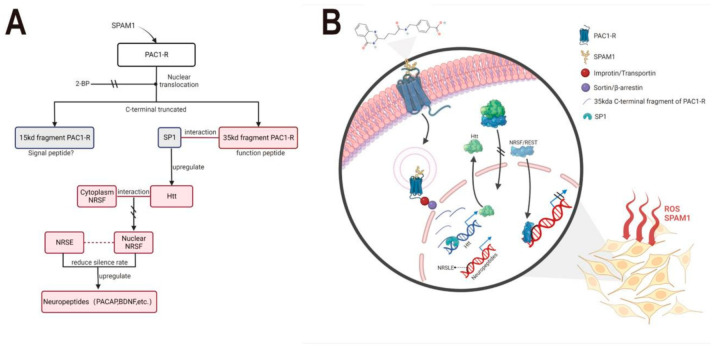
The hypothesis on the working mechanism of neuroprotective effect of SPAM1. (**A**) The flow chart of the mechanism of SPAM1 exerting neuroprotective effect; (**B**) the cartoon diagram of the mechanism of SPAM1 exerting neuroprotective effect.

**Table 1 ijms-23-15996-t001:** The amino acid residues involved in the binding with PAC1-R-EC1.

Small Molecules	Common Residues Binding to PAC1-R-EC1	Residues with Hydrogen-Bonding to PAC1-R-EC1
PACAP(30–37)	ASP24, ASP111, GLY114, PHE115, ASP116, GLU117, GLU119, GLU121, THR122	GLU121, ASP24, ASN37, LEU80, ASP111, GLY114, ASP116, TYR118, THR122
SPAM1	PHE27, ASP111, GLY114, PHE115, ASP116, GLU117, GLY123, ASP124	ASP116, GLY114, ASP124
TAT	ASP24, CYS25, PHE27, ASP111, PHE115, ASP116, GLU117, GLU119, GLU121, THR122	ASP24, ASP116, GLU117, GLU119, GLU121, GLN125, GLU359, SER120, ASP111
DOX	CYS25, PHE27, ASP111, GLY114, PHE115, ASP116	ASP116
Hydrazide	ASP24, ASP111, GLU117, SER120, GLU121 THR122, GLY123, ASP124	THR122

XXX, the common residues shared by four PAMs of PAC1-R; XXX, the common residues with hydrogen-bonding to PAC1-R-EC1 shared by PAMs of PAC1-R; XXX, the residues with two hydrogen-bonding to PAC1-R-EC1.

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
