# Peer review of "Novel Small Molecule Positive Allosteric Modulator SPAM1 Triggers the Nuclear Translocation of PAC1-R to Exert Neuroprotective Effects through Neuron-Restrictive Silencer Factor"

_ijms, 2022, doi:10.3390/ijms232415996_

Round 1

Reviewer 1 Report

This is a review of manuscript ijms-2093194 “SPAM1 triggers the nuclear translocation of PAC1-R to exert neuroprotective effects through neuronal-restrictive silencer factor” by Fan et al.

This study demonstrated that SPAM1 promotes the nuclear translocation of PAC1-R via negative regulation of NRSF nuclear translocation and protect neurons against oxidative stress.

This reviewer found this study is interesting, however, this reviewer has following minor concerns.

Miner points:

1. The title is not clear to represent this study. The authors should revise the title of this study.

2. Some abbreviations don’t spell out when they appeared at the first time in this manuscript, and H2O2 should be represented H2O2. In Results and Figures, figures and text do not much. The authors should check whole manuscript carefully.

3. In Fig.3, the order of graphs and figures is complicated. Could you be simpler?

4. In Fig.4A, the author should separate behavioral schedule or align orientation. The author mentioned that rats subjected to SPAM1 and PACAP38 treatment exhibited a significant decrease in damaged hippocampal neurons, but HE staining data is not enough to mention this. In addition, why don’t you show other regions of hippocampus?

5. There are mentioned rat, but the authors didn’t use rat.

6. The authors should mention clearly what mice first injected and second injected.

7. There are missing “saline”, the author should indicate saline if mice were injected with saline.

8.The authors should specify if what L, M, H-SPAM1 represent in figure legends.

9. In Fig.5D, mice injected PACAP38 seem like to be decreased in center region stay time.

10. In Fig.4, 5, and 6, the authors need to reconsider figure composition.

11. How about the effect of NAC and 2BP alone treatment on the nuclear translocation of NSRF and the production of neuropeptides? In addition, the authors can examine the effect of SPAM1 with H2O2 on the PACAP expression level and the nuclear translocation of PAC-1R.

11. In 2.10. “These findings suggest that increased levels of PAC1-R/PACAP are positively correlated with the nuclear translocation of PAC-1R.”, this reviewer could not understand this sentence.

12. In discussion, the author didn’t mention about behavioral tests. The author must mention about this, in particular that there are differences in locomotor activity in locomotor activity test sets, but not in Y-maze.

Reviewer 2 Report

In this paper, the authors reported that SPAM1 has a significant neuroprotective effect against oxidative stress in both H2O2-treated cells and D-galactose (D-gal) induced aging animal model. They described a novel mechanism by which SPAM exerts neuroprotective effects through inducing the nuclear translocation of PAC1-R and increasing the expression of nerve-related genes. The topic is of interest to readers and the findings are important. I have some minor comments listed below.

 1.   Long and convoluted sentences affect comprehension and readability. Please avoid too many long sentences in scientific writing.

2. Regarding the animal model, the description was confusing. Should it be D-gal-induced mice (e.g. line 187) or rats (e.g. line 182, 195)? Please clarify.

3.  Figure legends: generally the figure legend should include a title, the materials and methods involved with the presented figure, results (though this isn’t always applicable) and any other miscellaneous details such as explaining abbreviations or image scales. Please modify the legends accordingly.
